# OpenReview forum: "GASLITEing the Retrieval: Poisoning Knowledge DBs to Mislead Embedding-based Search"
_ICLR.cc/2025/Conference — Submitted to ICLR 2025_

### Official Review · Reviewer_dyPh · 2024-11-02

**Soundness:** 2
**Presentation:** 2
**Contribution:** 2
**Rating:** 5
**Confidence:** 5

**Summary:**

The authors propose a gradient-based corpus poisoning attack for dense retrieval systems. They consider 3 threat models that incrementally weaken the adversary's assumptions about query knowledge: "knows all", "knows little", "knows (almost) nothing". They use "appeared@k" as a metric to measure attack effectiveness, and show that their attack method outperforms two naive, and one state-of-the-art baseline. They provide further analysis about the distribution of the embedding space.

**Strengths:**

1) The related work section has very good coverage
2) There is a clear connection between each threat model examined, and the experiment that corresponds to it
3) In their experiments, the authors examine a variety of models

**Weaknesses:**

1) The novelty of the paper is quite low, and I find it difficult to understand what is the difference between the author's method, and HotFlip. To me it seems that the experiments under different threat models are closely resembling the experiments of HotFlip, where in every threat model we have a different definition of what a query is allowed to be.
2) The authors use non-standard terminology for standard concepts. For example, they use "embedding-based retrieval" for "dense retrieval", or "knowledge database" for "corpus". Using standard terminology is a way to emphasize the novelty of your contribution. Meaning that it is not best practice to make the reader think "this method has never been applied to knowledge databases before", when in reality your knowledge database is just the MSMARCO corpus.
3) In their motivation, the authors emphasize that it is not always the case that SEO attacks have access to user queries. However, all of their threat models have query assumptions -- even the "knows (almost) nothing" experiment, where we still have queries. I believe that a trully queryless attack would help drive your point forward.
4) The authors choose to present the result of each experiment differently, i.e., the "knows all" experiment corresponds to Table 1, the "knows what" experiment corresponds to Figs. 3 and 4 (which are given in the wrong order in the paper), and the "knows (almost) nothing" references Table 8 (which is in the appendix), and  Fig. 5. In reality, I do not have a solid foundation to compare the performance of their algorithm across different threat models.
5) In Table 1, the authors show that their method has 100% success rate for appeared@1, which means that their adversarially generated passages, always outrank relevant passages. This is an extremely strong result, that is not sufficiently explained and discussed. I would like to see some qualitative analysis about what the generated passages look like compared to the relevant passages, and further explanation about why would their method achieve this strong of a performance.
6) "The perfect attack" section does not really fit well with the rest of the story, it reads like an afterthought that was introduced in the paper way too late. Perhaps the authors need to introduce the concept earlier, and ease the reader into why this experiment corresponds to their research objectives. At this point, when we reach Section 7.1, it feels out of topic.

**Questions:**

Please, refer to my detailed concerns in the "Weaknesses" section.

**Details Of Ethics Concerns:**

The authors approach the topic of a corpus poisoning adversarial attack from the perspective of how to make the attack stronger, but do not discuss the perspective of how to make retrieval models more robust against it.

---

> ### Author Response · Authors · 2024-11-22
> **Response to Reviewer dyPh — Part 1/2**
>
> Thanks for the comments! We do our best to address these in the revision and in what follows, and are more than willing to answer any follow-up questions.
>
> > WK1. “The novelty of the paper is quite low, and I find it difficult to understand what is the difference between the author's method, and HotFlip.”
>
> Our primary contribution is exploring worst-case behavior of embedding-based retrievers, with GASLITE’s optimization method being a means toward doing so. We updated the revision to highlight this.
>
> GASLITE first introduces a _new_, interpretable centroid-based objective (Sec. 4.1, App. 2) and _critical_ improvements to the HotFlip-inspired optimization (Sec 4.2.) leading to significant increase in attack success (ablation studies in App. F). Indeed, HotFlip is a strong discrete optimization framework used in many attacks. However, **applying vanilla HotFlip or insufficiently improving it, can lead to a false sense of security**, due to low attack success (e.g., Sec. 6.2.).
>
> Evidently, GASLITE consistently _outperforms_ previous methods across threat models (Sec. 6) including HotFlip-based optimizers (Fig. 2) and matches a strong hypothetical attack enabled by our new objective (Sec 7.1).
>
> Crucially, offering GASLITE enables us to contribute novel key insights, including that _(1)_ white-box single-query attacks can achieve optimal success (Sec 6.1); _(2)_ cosine similarity leads to more robust retrievers (Sec. 6, Sec. 7.2); and _(3)_ embedding-space geometry correlates with robustness (Sec. 7.2). Importantly, GASLITE provides a reliable tool for evaluating defenses and the insights it enables set the groundwork for improving model robustness.
>
>
> > WK2. “non-standard terminology for standard concepts. For example, they use "embedding-based retrieval" for "dense retrieval", or "knowledge database" for "corpus".”
>
> We appreciate the reviewer’s attention to the terminology, and elaborate on our choices as follows. We also made clarification in the revision to avoid confusion; if reviewers see fit, we are open to making further changes.
>
> We choose _knowledge database_ over **_corpus_**, since:
>  - _corpus_ can be interpreted as the _training_ corpus, which might be misleading as our threat model does not assume (re-)training and as prior work refers to _Corpus Poisoning_ also in the context of poisoning _training_ data [1].
>  -  'corpus' traditionally connotes text-only data, while we refer to other modalities in the context of retrieval attacks (e.g., [2]).
>  -  ‘knowledge database poisoning’ is consistent with recent related attacks (e.g., [3, 4]).
>
> We use 'embedding-based retrieval' rather than 'dense retrieval' to precisely distinguish embedding-based bi-encoders—which we target—from other dense retrievers like cross-encoders (requiring query and passage to pass simultaneously in the model; [5]).
>
> > WK3. “authors emphasize that it is not always the case that SEO attacks have access to user queries...however, all of their threat models have query assumptions…”
>
> We clarify that only _Knows All_ has direct query access. Other threat models (_Knows What_, _Knows Nothing_) assume a varying knowledge of their _distribution_ (illustrated in Fig. 1). Crucially, our evaluation uses queries inaccessible to the attacker. We next discuss how the assumption about query-distribution knowledge could be further relaxed.
>
> >  WK3. “a truly queryless attack would help…”
>
> We thank the reviewer for this proposition, and combine existing and a new study to form a section on query-less attacks (App. H.5).  In App. H.5.1, we show that using synthetic (LLM-generated) queries it is possible to achieve comparable results under the _Knows What_ threat model. Additionally, the revision includes a cross-dataset transferability study (App. H.5.2): following Zhong et al.’s [6] results on successfully transferring their attack across datasets, we found GASLITE can leverage queries from one dataset (e.g., MSMARCO) to attack another datasets of different, entirely unknown query distributions.
>
> > WK4. “...not compare the performance of their algorithm across different threat models.”
>
> We thank the reviewer for the suggestion.  We now add Table 7 summarizing the results across threat models. Note that the comparison should be done while carefully considering the different assumptions of the threat models.
>
> A summary of the table, comparing GASLITE’s median appeared@10 (over models) across threat models:
>
> |_Knows All_ | _Knows What_|_Knows Nothing_|
> |---|---|---|
> |100% | 61.00% | 9.51%|
>
> As discussed in Sec. 7.2, less knowledgeable attackers tackle a more challenging setting.
>
> $>>>$

---

> > ### Author Response · Authors · 2024-11-22
> > **Response to Reviewer dyPh — Part 2/2**
> >
> > > WK5. “their method has 100% success rate for appeared@1 … This is an extremely strong result, that is not sufficiently explained and discussed”
> >
> > We thank the reviewer for recognizing this strong result—the revision discusses the result further (Sec. 6.1.) and includes adversarial passage examples (Table 8). In short, when dealing with single-query optimization, GASLITE achieves high vector similarity that suffices to surpass the similarity acquired by any other relevant passage (Table 8).
> >
> > >WK6. ”Perhaps the authors need to introduce the concept (perfect attack) earlier…”
> >
> > Thanks for the proposition. We agree that doing so would improve readability—the revision introduces the concept of the _perfect attack_ earlier (in Sec. 4.1).
> >
> > > Ethics. ”… perspective of how to make the attack stronger, but do not discuss the perspective of how to make retrieval models more robust”
> >
> > We extended the revision statement to discuss defenses. As demonstrated in past work [7], strong attacks can inform means to harden models. Additionally, in App. G. we evaluate several standard baseline defenses, including perplexity-filtering and outlier-detection based on L2-norm, both of which achieve varying levels of success against GASLITE (depending on the attack configuration).
> >
> > ### **References**
> >
> > [1] Schuster et al. 2020, Humpty Dumpty: Controlling Word Meanings via Corpus Poisoning
> >
> > [2] Zhou et al. 2020, Adversarial Ranking Attack and Defense
> >
> > [3] Zou et al. 2024, PoisonedRAG: Knowledge Corruption Attacks to Retrieval-Augmented Generation of Large Language Models
> >
> > [4] Shafran et al. 2024, Machine Against the RAG: Jamming Retrieval-Augmented Generation with Blocker Documents
> >
> > [5] Reimers and Gurevych 2019, Sentence-BERT: Sentence Embeddings using Siamese BERT-Networks
> >
> > [6] Zhong et al. 2023, Poisoning Retrieval Corpora by Injecting Adversarial Passages
> >
> > [7] Madry et al. 2018, Towards Deep Learning Models Resistant to Adversarial Attacks

---

> > ### Comment · Reviewer_dyPh · 2024-11-25
> >
> > Dear authors,
> > Thank you for a very extensive response. Upon review, I still have some objections about the issue of terminology:
> >
> > > corpus can be interpreted as the training corpus...
> >
> > In retrieval we typically have train/test queries, and the corpus remains constant. This is definitely true for collections such as MSMARCO and NQ that you use in your experimental setting.
> >
> > > 'corpus' traditionally connotes text-only data, while we refer to other modalities in the context of retrieval attacks
> >
> > The term "corpus" means "body", and it can refer to any type of collection regardless of its modality. Moreover, even if we accept that corpus traditionally refers to text-only data, in your experimental setup you do consider text-only data. You only refer to image retrieval in the related work section of Appendix A.
> >
> > > 'knowledge database poisoning' is consistent with recent related attacks (e.g., [3, 4])
> >
> > Both of these papers you cite are RAG papers, am I right to believe that your method targets retrieval systems, and not RAG? (sure, there is retrieval in RAG, but you do not talk about generation in your paper, so how can we assess the "knowledge" aspect?)
> >
> > Furthermore, I have a small objection about the additional experiments in Appendix H. These attacks are still not queryless. You still have queries, but they are synthetic. A queryless attack would be something along the lines of "perturb a document in the direction of another document", for example. But if your perturbation vector is grounded on a query, it is still not a queryless attack, even if the query is synthetic.

---

> > > ### Author Response · Authors · 2024-11-26
> > >
> > > We thank the reviewer for their time and detailed response.
> > >
> > > > **Terminology**
> > >
> > > We accept reviewer’s points and their suggestions regarding the term _corpus_. Our final revision will refer to text corpora with the term _corpus_ (including the work’s title).
> > >
> > > > **Queryless attacks**
> > >
> > > Thank you for clarifying the earlier suggestion. We agree that the attacks, including in App. H.5, utilize sample queries as part of their method. Note that, except for the Knows All setting (Sec. 6.1), the adversary’s samples are non-identical and semantically different from the test queries. App. H.5. further drops the assumption of _handing the attacker in-distribution queries_, by relying on commonly available alternatives (namely, LLM-generated queries or open-source datasets). For better clarity, in the final revision we will refer to these settings as the _synthetic query_ (App. H.5.1) and _surrogate query_ (App. H.5.2) settings.
> > >
> > > To our understanding, the reviewer suggests a method that, in lieu of using queries (even when these are readily available to attackers), produces adversarial passages using benign passages (=documents). For instance, to attack a concept such as iPhone, the adversary could sample iPhone-related passages from the corpus to create adversarial passages. Note that _this deviates from our threat model_, as we assume no access to the corpus which can be prohibitively large and dynamic (Sec. 3). In theory, however, the adversary could crawl relevant documents from the Internet or generate them synthetically (analogue to _synthetic query_ setting; App. H.5.1). We will discuss this direction in the next revision.

---

### Official Review · Reviewer_LdEb · 2024-11-04

**Soundness:** 2
**Presentation:** 3
**Contribution:** 2
**Rating:** 5
**Confidence:** 4

**Summary:**

This paper introduces the GASLITE, a poisoning attack targeting embedding-based text retrieval systems. The authors employ a gradient-based search method to craft adversarial passages, compelling the retriever to return information that the attacker intends to inject. Experimentally, GASLITE demonstrates significant superiority over other baselines across all settings, highlighting the heightened threat of poisoning attacks in text retrieval systems.

**Strengths:**

1. The authors conduct comprehensive experiments to validate the method's effectiveness. The experimental results indicate that GASLITE performs exceptionally well under three different knowledge conditions.
2. The writing is of high quality, and Figure 1 is clearly conveyed through the GASLITE workflow.
3. Given the widespread use of embedding-based text retrieval, focusing on its security is promising and meaningful.

**Weaknesses:**

1. My primary concern lies in the authors' assumptions. They assume that the attacker has white-box access to the embedding model, which significantly reduces the attack's practical applicability. Can the authors provide more real-world examples to support the hypothesis that attackers have such high-level access? Additionally, most current poisoning attacks assume black-box access for practical purposes. Can the authors explore how to implement poisoning attacks on text retrieval systems under black-box or gray-box conditions?

2. The method appears to be somewhat less innovative. How does a poisoning attack on text retrieval differ from other poisoning attacks on text data? Is the authors' method merely an adaptation of previous methods applied to text retrieval by modifying the attack algorithm's objective function? While I appreciate the list of improvements to the GASLITE algorithm in Section 4.2, these enhancements seem relatively minor.

3. The discussion of attacking text retrieval via KDB poisoning is too limited in the related work section. The authors should include more references that closely relate to their methodology. Additionally, the paper lacks discussion on defensive strategies and related experiments. For an emerging security issue, I am particularly interested in whether existing defense strategies can mitigate this risk.

4. Can the authors provide information on the time required to run the GASLITE algorithm? The authors mention in the limitations part that the white-box GASLITE algorithm is not very efficient, and constructing a black-box attack using zero-order gradient optimization may be more time-consuming. I would like to know whether attacking the text retrieval system in this context offers sufficient benefits.

**Questions:**

1. Design an attack algorithm under black-box or gray-box assumptions and provide the corresponding experimental results.
2. Include a discussion on attacking text retrieval via KDB poisoning and explore potential defense strategies.
3. Evaluate the time requirements of the attack method and assess its robustness when faced with defensive strategies.

---

> ### Author Response · Authors · 2024-11-22
> **Response to Reviewer LdEb — Part 1/2**
>
> Thanks for the comments! We do our best to address these in the revision and in what follows, and are more than willing to answer any follow-up questions.
>
> > WK1. “Can the authors provide more real-world examples to support the hypothesis that attackers have such high-level access?”
>
> A notable example is MeliSearch, a commercial search solution with [high-profile customers](https://www.meilisearch.com/customers) and open-source codebase that provides an “AI-powered search” that, by default, [uses](https://www.meilisearch.com/docs/reference/api/settings#embedders-object) the publicly available BGE-v1.5 embedding model. This model is similar to the ones we target. Another example is the popular [OpenInterpreter](https://www.openinterpreter.com/) app, that has code execution capabilities and uses open-source embedding (MiniLM-L6) for file search.
>
> The above examples emphasize the practical implications of our work. Still, we note that these practical implications should not overshadow key intellectual insights contributed by the paper, including the more precise characterization of retrievers’ susceptibility to KDB poisoning and that retrievers’ robustness is tied to the distribution of their representation space (specifically, models with more isotropic representations exhibit higher robustness; Sec. 7).
>
> > WK1, Q1. “Design an attack algorithm under black-box or gray-box assumptions and provide the corresponding experimental results”
>
> We thank the reviewer for the suggestion and agree that black-box attacks present an interesting avenue, especially given GASLITE’s strong results in white-box settings compared to established attacks. We next share the results included in the revision (Fig 21 in App. H.7).
>
> Following results in attack transferability [1, 2], we consider the 9 evaluated white-box attacks, and apply each one on target (“black-box”) models, we share here a notable slice of that evaluation (from Fig 21d; _Knows What_), each cell shows the attack’s _appeared@10_:
>
> |Source Model / Target Model |Arctic|Contriever|Contriever-MS|E5|aMPNet|mMPNet|
> |---|---|---|---|---|---|---|
> |**Arctic**|91.94|0|0|1.61|0|0|
> |**Contriever**|1.61|100|**62.9**|3.23|0|0|
> |**Contriever-MS**|1.61|**96.77**|100|0|0|0|
> |**E5**|**11.29**|6.45|0|80.65|1.61|0|0|0|
> |**aMPNet**|0|1.61|0|0|51.61|**4.84**|
> |**mMPNet**|0|0|0|1.61|**4.84**|67.74|
>
> We observe transferring attacks within model families or that share pretrained model (**bolded**) show different degrees of success. This, while GASLITE was _not_ explicitly optimized for transferability (e.g., via optimizing on multiple source models [2]), a direction for black-box attacks we intend to explore further in the future.
>
>
>
> > WK2. “The method appears to be somewhat less innovative. How does a poisoning attack on text retrieval differ from other poisoning attacks”
>
> Our primary contribution is exploring worst-case behavior of embedding-based retrievers, with GASLITE’s optimization method being a means toward doing so. We updated the revision to highlight this.
>
> GASLITE first introduces a _new_, interpretable centroid-based objective (Sec. 4.1, App. 2) and _critical_ improvements to the HotFlip-inspired optimization (Sec 4.2.) leading to significant increase in attack success (ablation studies in App. F). Indeed, HotFlip is a strong discrete optimization framework used in many attacks. However, **applying vanilla HotFlip or insufficiently improving it, can lead to a false sense of security**, due to low attack success (e.g., Sec. 6.2.).
>
> Evidently, GASLITE consistently _outperforms_ previous methods across threat models (Sec. 6) including HotFlip-based optimizers (Fig. 2) and matches a strong hypothetical attack enabled by our new objective (Sec 7.1).
>
> Crucially, offering GASLITE enables us to contribute novel key insights, including that _(1)_ white-box single-query attacks can achieve optimal success (Sec 6.1); _(2)_ cosine similarity leads to more robust retrievers (Sec. 6, Sec. 7.2); and _(3)_ embedding-space geometry correlates with robustness (Sec. 7.2). Importantly, GASLITE provides a reliable tool for evaluating defenses and the insights it enables set the groundwork for improving model robustness.
>
>
> > WK3. “authors should include more references that closely relate to their methodology”
>
> To our surprise we found little literature focusing on embedding-based retrieval KDB poisoning, with most prior attacks focused on cross-encoders retrievers (requiring query and passage to pass simultaneously in the model; [3]). Following the emergence of RAG, several recent attacks target this pipeline as a whole (albeit retrieval is not as extensively attended as in our work)—we already discuss past work in this domain (Sec. 2), and we will further discuss these, including the most recent work (e.g., [4, 5]) in the next revision. Of course, if the reviewers believe we should discuss other work, we will do our best to include it.
>
> $ >>> $

---

> ### Author Response · Authors · 2024-11-22
> **Response to Reviewer LdEb — Part 2/2**
>
> > WK4, Q3. “Evaluate the time requirements of the attack method…”
>
> Thanks for the suggestion; App. H.8 now reports detailed time requirements for different attacks and baselines. In a nutshell, gradient based attacks achieving non-negligible success have roughly the same time requirements (~1 single-GPU hour using
> 24GB VRAM for crafting a single passage) with GASLITE being significantly more successful (e.g., requiring 10 GPU hours for a successful concept-specific attack; Sec. 6.2). Baselines such as stuffing require negligible time, as they rely on simple string operations. However, these baselines also attain limited success.
>
>
> > Q2. “the paper lacks discussion on defensive strategies and related experiments”
>
> We explore possible defenses in App. G., evaluating several standard baseline defenses, including perplexity-filtering and outlier-detection based on L2-norm, both of which achieve varying levels of success against GASLITE (depending on the attack configuration). The revision now highlights this study in Sec. 6 and discusses defenses in the Ethics Statement.
>
>
> #### **References:**
> [1] Papernot et al. 2016, Transferability in Machine Learning: from Phenomena to Black-Box Attacks using Adversarial Samples
>
> [2] Zou et al. 2023, Universal and Transferable Adversarial Attacks on Aligned Language Models
>
> [3] Reimers and Gurevych 2019, Sentence-BERT: Sentence Embeddings using Siamese BERT-Networks
>
> [4] Chen et al. 2024, AgentPoison: Red-teaming LLM Agents via Poisoning Memory or Knowledge Bases
>
> [5] Pasquini et al. 2024, Neural Exec: Learning (and Learning from) Execution Triggers for Prompt Injection Attacks

---

> > ### Comment · Reviewer_LdEb · 2024-11-24
> >
> > I would like to thank the authors for their responses, which address some of my concerns, including the robustness of GASLITE under defensive strategies and the time requirements. However, my main concern remains unresolved. I understand that developing and implementing a black-box attack at the discussion stage is quite challenging. Nevertheless, based on the experiments provided by the authors, GASLITE appears to be largely ineffective in a black-box setting. In instances where there is a significant difference between the source model and the target model, GASLITE demonstrates a 0% success rate for the appeared@10 metric. Therefore, the authors' method, GASLITE, is experimentally applicable only to white-box settings.

---

> > > ### Author Response · Authors · 2024-11-26
> > >
> > > Thank you for the detailed response.
> > >
> > >
> > > We note that this work studies white-box attacks. Accordingly, GASLITE is _not_ optimized towards transferability in black-box settings, thus rendering the attack transferability within model families non-trivial. We offer and evaluate black-box baselines (e.g., _stuffing_ targeted queries is a strong black-box attack when a single targeted query is known; Sec. 6.1) and we agree the black-box setting is both interesting and valuable to explore in depth. We defer the design of attacks tailored for black-box settings to future work.
> > >
> > >
> > > **Why do white-box attacks matter?** We would like to highlight that white-box attacks such as GASLITE are crucial to explore, for multiple reasons:
> > > 1. White-box assumptions could be applicable to real-world applications (e.g., see the examples above).
> > > 2. Red team evaluations are often held in white-box settings (e.g., [1]), considering worst-case attacks.
> > > 3. These simulate worst-case attackers, following Kerchoff’s famous principle—the security of a system should not rely on obscurity (in our case, obscurity may refer to model weights, which could be leaked or extracted [2]).
> > > 4. Any successful attack (e.g., GASLITE) can provide insights into models’ vulnerabilities, such as the ones presented in Sec. 7. (e.g., our work is the first to relate the embedding-space isotropy to model robustness against poisoning).
> > > 5. As observed in past work (e.g., [3,4]), white-box attacks can serve as the basis of black-box attacks. One possible direction is following Zou et al. [4], and optimizing GASLITE’s objective against multiple architectures simultaneously. Next revision will highlight this in the conclusion.
> > >
> > > #### **References**
> > > [1] Mazeika et al. 2024, HarmBench: A Standardized Evaluation Framework for Automated Red Teaming and Robust Refusal
> > >
> > > [2] Jagielski et al. 2020, High Accuracy and High Fidelity Extraction of Neural Networks
> > >
> > > [3] Papernot et al. 2016, Transferability in Machine Learning: from Phenomena to Black-Box Attacks using Adversarial Samples
> > >
> > > [4] Zou et al. 2023, Universal and Transferable Adversarial Attacks on Aligned Language Models

---

> > > > ### Comment · Reviewer_LdEb · 2024-11-27
> > > >
> > > > Thank you for your further explanations. I agree with the authors' description of some use cases for white-box attacks. When evaluating the robustness of a model or a defense strategy, white-box attacks are often used to simulate worst-case scenarios. However, from the perspective of designing an attack method, white-box privileges undoubtedly reduce both the practicality and technical difficulty of the approach. Considering that GASLITE's main innovation lies in designing an attack method rather than a defensive strategy, and given its limited innovation compared to HotFlip, I maintain my rating.

---

> > > > > ### Author Response · Authors · 2024-11-28
> > > > >
> > > > > Thank you for the response.
> > > > >
> > > > > We would only like to clarify that while GASLITE’s white-box optimization method is one contribution, the _primary_ contributions lie in the _extensive evaluation under the more realistic threat models we propose and our novel findings enabled by GASLITE_, including that: _(1)_ single-query attacks can achieve optimal success (Sec. 6.1); _(2)_ dot-product models are more susceptible to poisoning attacks both mathematically and empirically (Sec. 7.2); and _(3)_  models’ robustness to poisoning attacks is related to their representation spaces’ isotropy (Sec. 7.2).
> > > > >
> > > > > We apologize if these contributions were underemphasized in the original submission. _We have attempted to clarify them in the current revision and will further highlight them in future revisions_. Of course, we understand if the reviewer wishes to maintain their score and thank them again for their time and valuable feedback.

---

### Official Review · Reviewer_pceo · 2024-11-04

**Soundness:** 3
**Presentation:** 3
**Contribution:** 3
**Rating:** 6
**Confidence:** 3

**Summary:**

This paper discusses the vulnerability of embedding-based text retrieval systems to search-engine optimization (SEO) attacks. The authors propose a gradient-based attack method called GASLITE, which generates adversarial passages to be inserted into knowledge databases (KDBs) in order to manipulate search results for specific queries or concepts. The attack aims to promote malicious content by ensuring that the adversarial passages rank highly for targeted query distributions without relying on the KDB content or modifying the retrieval model.

**Strengths:**

1. No need to update model parameters/additional training.
2. No need for read permission of KDB.
The method is better than the baseline method in both Knows All and Knows What scenarios.
At the same time, |Padv| << |P|, and the position rate is less than or equal to 0.0001%.
A small number of samples achieve the maximum SEO attack effect.
It also analyzes various factors that affect the GASLITE attack method in this paper: similarity measurement methods, vector space characteristics of the Embedding model, diversity distribution of target Queries.

**Weaknesses:**

1. Computational Intensity: The GASLITE attack method is computationally intensive, especially for LLM-based embedding retriever, requiring significant resources, which may limit its scalability and practical application in certain scenarios. This computational demand could be a barrier for less resource-intensive environments or smaller-scale attacks.
2. Dependence on External Data: The effectiveness of the attack relies on the ability to generate or obtain a sample of queries related to the targeted concept. If the attacker cannot generate or obtain a representative sample, the attack's effectiveness may be diminished.
3. Impact of Tokenization: The paper acknowledges that the tokenization process can affect the attack's success, as certain token lists cannot be reached from any text. This implies that the attack's effectiveness is tied to the specific tokenizer used by the embedding model.

**Questions:**

1. Why didn't embedding-based retrievers use models such as bge-en-icl 7kM, stella_en_1.5B_v5(1kM), gte-Qwen2-7B-instruct(7kM), and GritLM-7B(7kM)?
2. How about the effect of these Embedding models with parameters of around 7000M?
3. Do you think there is a better form than info prefix + trigger for Padv?
4. How is the metric informativeness measured?
5. What if the embedding-based retriever is a black box model and we cannot obtain its weights parameters, how can we conduct this SEO attack ?

---

> ### Author Response · Authors · 2024-11-22
> **Response to Reviewer pceo — Part 1/2**
>
> Thanks for the comments! We do our best to address these in the revision and in what follows, and are more than willing to answer any follow-up questions.
>
> > WK1. “computational demand could be a barrier for less resource-intensive environments or smaller-scale attacks.”
>
> In our study, we focus on retrievers’ susceptibility to _worst-case_ attacks (e.g., a compute-abundant attacker; Sec. 3), which are also critical for revealing inherent model weaknesses (as done in Sec. 7.2) and can enable “red teaming” to assess future defenses. Finally, GASLITE shows higher success rates in shorter runtime compared to previous attacks (Fig. 2). The revision includes more details on resource usage (App. H.6).
>
> > WK2. “If the attacker cannot generate or obtain a representative sample, the attack's effectiveness may be diminished”
>
> We thank the reviewer for raising this concern, and include the next study in the revision (App. H.5.2).
>
> Following the findings of Zhong et al. [1], we show that GASLITE can be used to craft adversarial passages from one dataset (e.g., MSMARCO)—non-representative of the target queries—that successfully attack another target dataset (e.g., FiQA). The following table summarizes the appeared@10 achieved in each variant:
>
> |Source Queries / Target Queries|msmarco|nq|quora|scifact|fiqa|
> |---|---|---|---|---|---|
> |msmarco|9.5%|20.5%|9.4%|82.0%|30.5%|
>
> This shows the possibility of transferring attacks crafted using available, _source_ queries to _target_ queries of entirely different distribution. Topic-specific datasets (FIQA and SciFact) show high susceptibility to this method. Moreover, _source_ queries can also be synthetic (LLM-generated), as demonstrated in App. H.5.1.
>
> > WK3. “tokenization process can affect the attack's success”
>
> Through our evaluation we attack at least 4 different tokenizers (BERT’s, T5’s,  MPNet’s,  and Qwen2’s). Empirically, we do not find that tokenizers affect attack success.
>
> > Q1, Q2. “Why didn't embedding-based retrievers use models such as bge-en-icl 7kM, stella_en_1.5B_v5(1kM), …?”
>
> We thank the reviewer for the suggestion and agree LLM-based retrievers are interesting to explore, albeit not our focus in the paper (Sec. 5). We are currently working on and will add a detailed attack evaluation of `stella_en_1.5B_v5` to the next revision (App. H.4), and will do our best to target a 7B LLM-based as well.
>
> > Q3. “Do you think there is a better form than info prefix + trigger for Padv?”
>
> We have tested different trigger locations (suffix / prefix / middle) (App. F). In terms of attack _performance_, placing the trigger as a prefix presents a stronger attack. However, targeting retrieval as part of a realistic attack (e.g., spreading misinformation in search, or injecting a prompt to RAG), we opt the trigger to serve as a suffix. This way, for example, humans exposed to the adversarial passage through search would first read to the misinformation, providing a more effective exposure per the [Primacy Effect](https://link.springer.com/referenceworkentry/10.1007/978-1-4419-1698-3_1116).
>
> > Q4. “How is the metric informativeness measured?”
>
> Per the threat model (Sec. 3), we view informativeness as a binary objective—an attack either contains malicious information or not. Since in all experiments the informative prefix is included in the adversarial passages, all attacks satisfy this goal by design (Sec. 5).
>
> $>>>$

---

> > ### Author Response · Authors · 2024-11-22
> > **Response to Reviewer pceo — Part 2/2**
> >
> > > Q5. “What if the embedding-based retriever is a black box model and we cannot obtain its weights parameters”
> >
> > We thank the reviewer for the suggestion and agree black-box setting is interesting to explore. We next share the results included in the revision (Fig 21 in App. H.7).
> >
> > Following results in attack transferability [2, 3], we consider the 9 evaluated white-box attacks, and apply each one on target (“black-box”) models, we share here a notable slice of that evaluation (from Fig 21d; _Knows What_), each cell shows the attack’s _appeared@10_:
> >
> > |Source Model / Target Model |Arctic|Contriever|Contriever-MS|E5|aMPNet|mMPNet|
> > |---|---|---|---|---|---|---|
> > |**Arctic**|91.94|0|0|1.61|0|0|
> > |**Contriever**|1.61|100|**62.9**|3.23|0|0|
> > |**Contriever-MS**|1.61|**96.77**|100|0|0|0|
> > |**E5**|**11.29**|6.45|0|80.65|1.61|0|0|0|
> > |**aMPNet**|0|1.61|0|0|51.61|**4.84**|
> > |**mMPNet**|0|0|0|1.61|**4.84**|67.74|
> >
> > We observe transferring attacks within model families or that share pretrained model (**bolded**) show different degrees of success. This, while GASLITE was _not_ explicitly optimized for transferability (e.g., via optimizing on multiple source models [3]), a direction for black-box attacks we intend to explore further in the future.
> >
> > #### **References**:
> > [1] Zhong et al. 2023, Poisoning Retrieval Corpora by Injecting Adversarial Passages
> >
> > [2] Papernot et al. 2016, Transferability in Machine Learning: from Phenomena to Black-Box Attacks using Adversarial Samples
> >
> > [3] Zou et al. 2023, Universal and Transferable Adversarial Attacks on Aligned Language Models

---

> ### Author Response · Authors · 2024-11-28
>
> Thank you, once again, for the constructive reviews. We have made our best to address the reviewer feedback in the latest revision. Of course, if any additional questions remain, we would be happy to answer them.
>
> If the reviewers believe we have satisfactorily addressed their concerns, we would kindly request their reconsideration of the rating.

---

### Official Review · Reviewer_NUva · 2024-11-11

**Soundness:** 3
**Presentation:** 3
**Contribution:** 3
**Rating:** 8
**Confidence:** 3

**Summary:**

The paper introduces GASLITE, a novel adversarial attack strategy designed to exploit vulnerabilities in embedding-based text retrieval systems. By inserting adversarial passages into knowledge databases (KDBs), GASLITE aims to manipulate search results to favor malicious content being retrieved in the top results. The method employs a gradient-based search to craft passages that not only embed adversary-chosen information but also achieve high visibility in search results for a given query distribution when added to KDBs.

Key contributions include:
1. GASLITE effectively targets the practical challenges of embedding-based retrieval vulnerabilities, focusing on search-engine optimization (SEO) attacks, which aim to present adversarial content prominently in search results.
2. A detailed analysis of threat models and attacker capabilities, which provides a robust framework for understanding and assessing the vulnerabilities in embedding-based search systems.
3. The study conducts extensive empirical evaluations across multiple state-of-the-art retrieval models, demonstrating GASLITE's superior performance compared to previous methods at minimal poison rates. It also considers several baselines and novel threat models closely aligned with SEO goals.

**Strengths:**

1. The paper is generally well-written, with clear explanations of complex concepts such as the gradient-based optimization process.
2. The experimental design is robust, utilizing a diverse array of retrieval models and demonstrating GASLITE's efficacy across different scenarios. This thorough evaluation enhances the credibility of the results and underscores the method's practical relevance.
3. The paper demonstrates high success rates of the GASLITE attack across multiple retrieval models, achieving impressive results even at low poison rates. These results when compared to baselines are quite impressive and move the filed forward. The flexibility to add further constraints like fluency further demonstrates the effectiveness of the method.

**Weaknesses:**

1. The assumption of white-box access, while useful for theoretical exploration, might not fully align with real-world scenarios where such access is restricted. Discussing alternative practical attack vectors with limited access would strengthen the work's applicability insights. Most search engines of interest has proprietary models and the knowledge about open source models used in not available. It would be interesting to analyze how the poison passages found using one model generalizes to others.
2. While the paper convincingly demonstrates GASLITE's effectiveness, it lacks a detailed discussion on the method's scalability across larger datasets and systems, a factor crucial for understanding its operational feasibility in extensive real-world applications. Even when comparing with baselines, the runtimes of the systems are not compared. Often than not, the attackers have limited compute budget as well and hence discussing that aspect would have been helpful.

**Questions:**

1. Can you provide some analysis and comparison of time and resources required to perform these attacks and compare the same for the baselines?
2. Can you share more examples of the generated passages for other concepts than potter and sandwich, especially the ones with fluency constraints?
3. How does the method compare with summarizing the top 10 results for all the queries in the set. The examples on Potter looked like it generated main/popular keywords about Potter in the generated doc to boost its retrieval probability across the common queries.
4. In your selection of the seen and unseen queries for evaluation, do you take care of semantic similarity? If there is a huge overlap of semantic similarity between the train and test query set, it may not be that representative of the threat model.

---

> ### Author Response · Authors · 2024-11-22
> **Response to Reviewer NUva — Part 1/2**
>
> Thanks for the comments! We do our best to address these in the revision and in what follows, and are more than willing to answer any follow-up questions.
>
> > WK1. “It would be interesting to analyze how the poison passages found using one model generalizes to others.”
>
> We thank the reviewer for the suggestion and find such a question interesting as well. We updated the revision to include a study on the transferability of GASLITE attacks.
>
> Following results in attack transferability [1, 2], we consider the 9 evaluated white-box attacks, and apply each one on target (“black-box”) models, we share here a notable slice of that evaluation (from Fig 21d; _Knows What_), each cell shows the attack’s _appeared@10_:
>
> |Source Model / Target Model |Arctic|Contriever|Contriever-MS|E5|aMPNet|mMPNet|
> |---|---|---|---|---|---|---|
> |**Arctic**|91.94|0|0|1.61|0|0|
> |**Contriever**|1.61|100|**62.9**|3.23|0|0|
> |**Contriever-MS**|1.61|**96.77**|100|0|0|0|
> |**E5**|**11.29**|6.45|0|80.65|1.61|0|0|0|
> |**aMPNet**|0|1.61|0|0|51.61|**4.84**|
> |**mMPNet**|0|0|0|1.61|**4.84**|67.74|
>
> We observe transferring attacks within model families or that share pretrained model (**bolded**) show different degrees of success. This, while GASLITE was _not_ explicitly optimized for transferability (e.g., via optimizing on multiple source models [2]), a direction for black-box attacks we intend to explore further in the future.
>
> Still, we note that practical implications of GASLITE should not overshadow key intellectual insights contributed by the paper, including the more precise characterization of retrievers’ susceptibility to KDB poisoning and that retrievers’ robustness is tied to the distribution of their representation space (specifically, models with more isotropic representations exhibit higher robustness; Sec. 7).
>
> > Wk2, Q1. “the attackers have limited compute budget as well and hence discussing that aspect would have been helpful…provide some analysis and comparison of time and resources”
>
> We thank the reviewer for their suggestion; App. H.8 now reports detailed time requirements for different attacks and baselines. In a nutshell, gradient based attacks achieving non-negligible success have roughly the same time requirements (~1 single-GPU hour using $<$24GB VRAM for crafting a single passage) with GASLITE being significantly more successful (e.g., requiring 10 GPU hours for a successful concept-specific attack; Sec. 6.2). Baselines such as _stuffing_ require negligible time, as they rely on simple string operations. However, these baselines also attain limited success.
>
> > Q2. “share more examples of the generated passages for other concepts”
>
> The revision includes an additional table with concept-specific attack examples (Table 11).
>
> > Q3. “How does the method compare with summarizing the top 10 results for all the queries in the set”
>
> We thank the reviewer for this interesting suggestion for an additional baseline, and include it in the revision (App. H.6). We note that we do _not_ assume an attacker's access to knowledge base, hence this baseline deviates from our threat model. Still, we find value in evaluating this attack with the more permissive assumptions.
>
> In an initial attempt to apply this idea we take a specific concept, and for each cluster of queries (out of 10) we ask Claude-Sonnet-3.5 to craft a summarized paragraph, which we use as the adversarial suffix. We repeat evaluation done in Sec. 6.2, and compare to other attacks:
>
> |Concept|Info Only|Query Stuffing|Top-Passages-Sum|GASLITE|
> |---|---|---|---|---|
> |**Mortgage**|0%|1%|2%|72%|
> |**Potter**|0%|5%|18%|81%|
>
> Results on the Potter concept seem to affirm the reviewer’s hypothesis—the proposed attack (Top-Passage-Sum) is more effective than the _Info Only_ and _Query-Stuffing_ baselines, still, it is markedly outperformed by GASLITE.
>
> $>>>$

---

> > ### Author Response · Authors · 2024-11-22
> > **Response to Reviewer NUva — Part 2/2**
> >
> > > Q4. “If there is a huge overlap of semantic similarity between the train and test query set, it may not be that representative of the threat model”
> >
> >
> > We thank the reviewer for raising this concern. Through additional experiments (added to revision; App. E.4, H.2), we find that _(1)_ extreme semantic overlap between our randomly split attack (=train) and eval (=test) query sets (e.g., synonymous pairs), is rare to none existent in our evaluation (Fig 11; App. E.4.), and _(2)_ even evaluating on a harder portion of queries, _most_ distant from the attack distribution (can be seen as discarding the more popular queries, and keeping ‘niche' queries), provide similar trends to the results reported in the original submission (Tab. 9; H.2.).
> >
> >
> > Additionally, in a new study in App H.5.2, following the findings of Zhong et al. [3], we show that GASLITE can be used to craft adversarial passages from one dataset (e.g., MSMARCO)—non-representative of the target queries—that successfully attack another target dataset of different query distributions (e.g., FiQA, NQ).
> >
> > #### **References:**
> > [1] Papernot et al. 2016, Transferability in Machine Learning: from Phenomena to Black-Box Attacks using Adversarial Samples
> >
> > [2] Zou et al. 2023, Universal and Transferable Adversarial Attacks on Aligned Language Models
> >
> > [3] Zhong et al. 2023, Poisoning Retrieval Corpora by Injecting Adversarial Passages

---

> ### Author Response · Authors · 2024-11-28
>
> Thank you, once again, for the constructive reviews. We have made our best to address the reviewer feedback in the latest revision. Of course, if any additional questions remain, we would be happy to answer them.
>
> If the reviewers believe we have satisfactorily addressed their concerns, we would kindly request their reconsideration of the rating.

---

> ### Comment · Reviewer_NUva · 2024-12-01
> **Response to the rebuttal**
>
> I would like to thanks the authors for responding to the questions and weaknesses pointed out in the review.
>
> The additional examples and the summarization baseline shows that the model is able to exploit the embedding space and its peculiarities.
>
> It would be helpful for the readers if the authors can dissect and analyze the model results more to understand what makes it work. (Just a helpful suggestion as a reader with no impact to my rating)
>
> I have revised my rating in view of the results.
>
> Thanks and best of luck.

---

### Author Response · Authors · 2024-11-22
**General Response to All Reviewers**

We sincerely thank reviewers for their constructive comments and suggestions! We did our best to address these in the **revised PDF**, and will continue to do so in the following ones. For the reviewers’ convenience, the we **highlighted the main updates** (in yellow).

**Major additions/changes include:**
- _Challenging query-set_ evaluation under Knows What, i.e., concept attacks (App. H.2)..
- _Queryless_ attacks, including the new study on cross-datasets transferability (App H.5).
- Comparison with the suggested _summarization baseline_ (App. H.6).
- _Black-box evaluation_ through cross-model transferability study, with even more results to be included in the next revision (App. H.7).
- _Run-time comparison_ between attacks and baselines (App. H.8).
- Discussing _potential mitigations_ in the ethics statement (in addition to the evaluation previously included).
- More _attack examples_: Knows All (=single queries attacks) samples in Table 8 and Knows What in Table 11.
- Addressing other issues and making clarifications, per reviewer suggestions.

**We plan future revisions to include:**
- Attacking LLM-based retrieval (e.g., Stella-1.5B).
- More detailed discussion of RAG attacks related work.

---

### Meta-Review · Area_Chair_24c7 · 2024-12-17

**Metareview:**

This paper presents a framework for attacking retrieval systems. The authors make certain assumptions about the embedding models and query distributions used in these systems, and their experiments demonstrate strong results under these specific conditions.

Strengths:
- The experimental results are impressive.
- The paper provides a comprehensive overview of related work.
- The paper is well-written and the figures are clear and easy to understand.

Weaknesses:
- The technical novelty of the proposed method is limited.
- The assumptions made about the query distributions and embedding models are unrealistic for real-world scenarios, limiting the practical applicability of the work.
- The high computational cost of the attacks further reduces their practicality.

Despite its strengths, the paper's limitations in terms of novelty and practical relevance lead me to recommend rejection.

**Additional Comments On Reviewer Discussion:**

The authors made a commendable effort to address the reviewers' concerns through additional experiments and clarifications. While they successfully resolved some issues, critical concerns regarding the underlying assumptions and the method's practical applicability remain. If the authors can further develop this work to address these limitations, the paper's impact and relevance would be significantly enhanced.

---

### Decision · Program_Chairs · 2025-01-22

Reject